# A prediction model for COVID-19 liver dysfunction in patients with normal hepatic biochemical parameters

Jianfeng Bao[1,*], Shourong Liu[1,*], Xiao Liang[2,3,4,*] , Congcong Wang[5], Lili Cao[6], Zhaoyi Li[5], Furong Wei[5], Ai Fu[5], Yingqiu Shi[2,3,4], Bo Shen[7], Xiaoli Zhu[7], Yuge Zhao[8], Hong Liu[8], Liangbin Miao[5], Yi Wang[5], Shuang Liang[2,3,4], Linyan Wu[6], Jinsong Huang[1] , Tiannan Guo[2,3,4] , Fang Liu[5]

Coronavirus disease 2019 (COVID-19) patients with liver dysfunction (LD) have a higher chance of developing severe and critical disease. The routine hepatic biochemical parameters ALT, AST, GGT, and TBIL have limitations in reflecting COVID-19–related LD. In this study, we performed proteomic analysis on 397 serum samples from 98 COVID-19 patients to identify new biomarkers for LD. We then established 19 simple machine learning models using proteomic measurements and clinical variables to predict LD in a development cohort of 74 COVID-19 patients with normal hepatic biochemical parameters. The model based on the biomarker ANGL3 and sex (AS) exhibited the best discrimination (time-dependent AUCs: 0.60–0.80), calibration, and net benefit in the development cohort, and the accuracy of this model was 69.0–73.8% in an independent cohort. The AS model exhibits great potential in supporting optimization of therapeutic strategies for COVID-19 patients with a high risk of LD. This model is publicly available at https://xixihospital-liufang.shinyapps.io/DynNomapp/.

## Introduction

Coronavirus disease 2019 (COVID-19), which is caused by severe acute respiratory syndrome coronavirus 2 (SARS-CoV-2), has become a pandemic. Different levels of liver disease have been reported in more than half of the infected population (Cai et al, 2020; Jothimani et al, 2020) and are recognized as an essential component of COVID-19 (Marjot et al, 2021).

In the early stage of COVID-19, some patients with normal hepatic biochemical parameters (alanine aminotransferase, aspartate aminotransferase, γ-glutamyl transferase, and total bilirubin) experience progression to liver dysfunction (LD) after the initiation of hepatotoxic-antiviral drugs, with even dramatic progression to liver failure without predictable circumstances (Carothers et al, 2020; Weber et al, 2020). With the reduction in the rate of COVID-19–related severity and death, a means of decreasing the incidence of non-SARS-CoV-2–related complications has become increasingly important for patients with mild disease.

Given the important role of the liver in drug metabolism, coagulation, albumin synthesis, and the generation of acute-phase reactants, abnormal liver function can influence the pathologic course of the systemic disease COVID-19. Accumulating evidence suggests that elevated expression of serum hepatic biochemical parameters is related to adverse events, including mechanical ventilation, shock, and increased risk of death (Cai et al, 2020; Ding et al, 2021; Mao et al, 2020; Weber et al, 2021; Yadav et al, 2021; Yip et al, 2021). However, hepatic abnormalities involve various manifestations, including liver congestion, inflammatory response, drug-induced liver damage, and hepatocyte infection (Marjot et al, 2021), with associated difficulty in identifying susceptible patients at an early stage. Considering the limitations of common hepatic biochemical parameters in representing the actual liver function status of COVID-19 patients, early identification of subgroups with a high risk of COVID-19–based LD through new biomarkers is needed for medical intervention. However, there is no reliable strategy thus far for effective prediction of subsequent LD among COVID-19 patients at admission.

Proteomics profiling has the ability to shed light on molecular changes reflected in sera from COVID-19 patients (Nie et al, 2021; Shen et al, 2020). In this study, we developed and validated an exceptionally parsimonious model through proteomic analysis to predict subsequent risk of liver damage among COVID-19 patients with normal hepatic biochemical parameters on admission, in

[1]Department of Hepatology, Affiliated Hangzhou Xixi Hospital, Zhejiang University School of Medicine, Hangzhou, China  [2]Westlake Laboratory of Life Sciences and Biomedicine, Key Laboratory of Structural Biology of Zhejiang Province, School of Life Sciences, Westlake University, Hangzhou, China  [3]Institute of Basic Medical Sciences, Westlake Institute for Advanced Study, Hangzhou, China  [4]Center for Infectious Disease Research, Westlake Laboratory of Life Sciences and Biomedicine, Hangzhou, China  [5]Insititute of Hepatology and Epidemiology, Affiliated Hangzhou Xixi Hospital, Zhejiang University School of Medicine, Hangzhou, China  [6]Department of Nursing, Affiliated Hangzhou Xixi Hospital, Zhejiang University School of Medicine, Hangzhou, China  [7]Department of Laboratory Medicine, Taizhou Hospital of Zhejiang Province Affiliated to Wenzhou Medical University, Linhai, China  [8]Department of Pathology, Affiliated Hangzhou Xixi Hospital, Zhejiang University School of Medicine, Hangzhou, China

Correspondence: huangjinsongyz@126.com; guotiannan@westlake.edu.cn; hzxxlf@outlook.com
*Jianfeng Bao, Shourong Liu, and Xiao Liang contributed equally to this work.

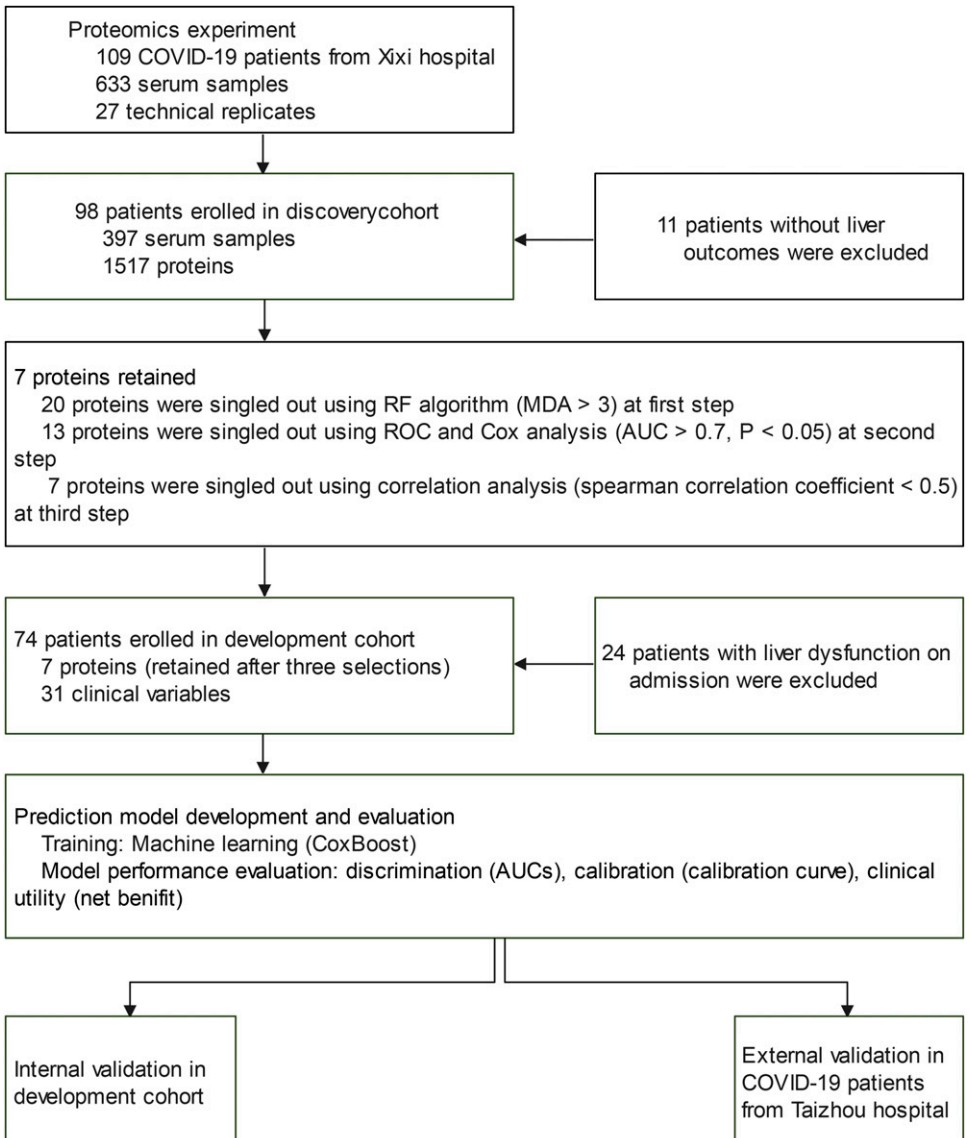

**Figure 1.  Study design and modeling flowchart.**
RF, random forest; MDA, the mean decrease of accuracy; ROC, receiver operator characteristic curve.

accordance with Transparent Reporting of a multivariable prediction model for Individual Prognosis Or Diagnosis (TRIPOD) standards.

Currently, many clinical predictive models might not have wide adaptability in clinical practice, often because of modeling of complex and multiple variables. In addition, most such models have been classified as having a high risk of bias and might not be generalizable, which is often caused by nonadherence to reporting standards and best practice methods during model development (Wolff et al, 2019; Wynants et al, 2020). The model reported here includes only two variables, namely, sex and angiopoietin-like 3 (ANGL3), which is a liver-specific secreted protein (Gaudet et al, 2017) screened by random forest (RF) and CoxBoost algorithm analyses from 1,517 proteins of proteomic data. The generalizability concerning discrimination, calibration, and clinical utility of this prediction model was comprehensively evaluated according to

TRIPOD guidelines. We call this the protein-based AS LD model. The prediction tool and model parameters are freely available online to facilitate risk stratification for therapeutic interventions.

# Results

### Baseline clinical characteristics of the discovery cohort

As shown in Fig 1, 109 participants were recruited between January 2020 and April 2020. During this period, 660 samples were collected from all of these patients. Ninety-eight (89.9%) individuals with available liver outcomes were included in the discovery cohort. A total of 397 serum samples from these individuals were designated as a source of material for identifying biomarkers. The baseline characteristics of the discovery cohort are shown as stratification

for LD (Table S1); 52 (53.1%) of the patients experienced LD, with a median follow-up time of 19 (median value and interquartile range [IQR]: 10.2–31.0) days. Significant differences were observed in sex and body mass index (BMI) ($P < 0.05$).

## Protein selection

Proteomic analysis identified 1,517 proteins from the serum samples. 20 proteins were preliminarily screened by a RF algorithm with a mean decrease in accuracy (MDA) of higher than 3 (Fig S1). The expression patterns of these 20 proteins among different grades of LD samples were compared using principal coordinates analysis (PCoA) mapping (Fig S2). The first and second principal components (PCoA1 accounted for 43.16% of the variance and PCoA2 for 11.01%) were plotted using the proteomic data, revealing a certain degree of distinction among the four groups.

A heat map was plotted to illustrate the expression patterns of the above 20 proteins in different grades of LD samples (Fig 2A). AL1A1 was first removed owing to a negligible difference between the LD groups. Thirteen proteins were retained according to the following rules: AUC > 0.70 for any grade of LD (Fig 2B) and significant multivariable associations with LD ($P < 0.05$) (Fig 2C). To minimize potential collinearity and overfitting for variables in the final model, we removed six proteins (FAAA, PSA7, PSB1, PSA5, PSA3, and PSA1) that correlated strongly with ALDOB and PSA4 (Spearman correlation coefficient > 0.6) (Fig 2D).

Levels of the retained proteins (ANGL3, ALDOB, ADH4, ACY1, ADH1B, ARLY, and PSA4) were significantly different between the normal and LD groups (Fig 3A). Patients in the LD group exhibited higher levels of ANGL3 (median IQR: 1.06 [1.27–0.95] versus 1.03 [1.11–0.89], $P < 0.01$), ALDOB (median IQR: 1.52 [1.91–1.07] versus 0.84 [1.02–0.68], $P < 0.001$), ADH4 (median IQR: 0.59 [1.11–0.59] versus 0.59 [0.59–0.41], $P < 0.001$), ACY1 (median IQR: 0.56 [0.79–0.56] versus 0.56 [0.56–0.43], $P < 0.001$), ADH1B (median IQR: 0.99 [1.50–0.82] versus 0.79 [0.82–0.51], $P < 0.001$), ARLY (median IQR: 0.75 [1.02–0.75] versus 0.75 [0.75–0.61], $P < 0.001$), and PSA4 (median IQR: 0.96 [1.32–0.85] versus 0.81 [0.90–0.68], $P < 0.001$).

## Baseline characteristics of the development cohort

In the development cohort, 30 patients (40.5%) experienced LD within a median follow-up of 22.0 (11.0–32.0) days (Table 1). No significant differences were observed between the patients with normal liver function and those who experienced LD during the follow-up period.

## Predictor selection

Baseline measurements included 31 clinical variables at admission and peak values of seven prioritized proteins within 7 d of hospitalization, which were simultaneously introduced into the Cox-Boost model for predictor selection. After CoxBoost processing, 10 variables remained as significant predictors of LD, including ADH4, hypertension, sex, fever, ANGL3, cough, feebleness, smoking, Hepatis B virus (HBV), and chest computed tomography (CT) (Fig S3). The step number and the penalty of the CoxBoost classifier were 101 and 198, respectively.

## Model selection

To maximize the clinical convenience and applicability of our prediction model, we limited the predictors to two in the final model and constructed 19 models as follows: ANGL3.Sex (ANGL3.SE), ANGL3.HBV (ANGL3.HB), ANGL3.Hypertension (ANGL3.HY), ANGL3.S-moking (ANGL3.SM), ANGL3.Chest CT (ANGL3.CT), ANGL3.Fever (ANGL3.FEV), ANGL3.Cough (ANGL3.CO), ANGL3.Feebleness (ANGL3.-FEE), ANGL3.ADH4, ANGL3, ADH4.Sex (ADH4.SE), ADH4.HBV (ADH4.HB), ADH4.Hypertension (ADH4.HY), ADH4.Smoking (ADH4.SM), ADH4.Chest CT (ADH4.CT), ADH4.Fever (ADH4.FEV), ADH4.Cough (ADH4.CO), ADH4.-Feebleness (ADH4.FEE), and ADH4 (Fig 3C).

To assess the model performance over the entire period, we calculated the time-dependent areas under the curves (AUCs) and presented them by graphical visualization (Figs 3B and S4). The ANGL3-based and ADH4-based models yielded AUC values from 0.48–0.83 to 0.53–0.81 over time, respectively. The calibration curves of ANGL3-based models showed better performance than those of ADH4-based models (Figs 3B and S5). The discrimination and calibration of the model ADH4 could not be calculated.

Decision curve analyses of the 19 models at a point estimate of 28 d are shown in Fig 3C. Threshold probabilities for the net benefit associated with the application of the ANGL3-based and ADH4-based models in detecting LD ranged from 0.00–0.78 and 0.00–0.46, respectively. Notably, the ANGL3.SE (AS) model showed a higher net benefit than any other ANGL3-based model and the treat all or treat none strategies.

After performing an omni-ensemble analysis with discrimination, calibration, and clinical utility assessment, we prioritized the best-of-state AS model as the final LD prediction model for COVID-19 patients. As shown in Fig 3B, the time-dependent AUCs were 0.60–0.80 for predicting LD, indicating favorable discrimination by the AS model. The calibration curves of the AS model showed adequate agreement between the predicted and observed probability in the development cohort. Decision curves revealed that the AS model achieved more net benefits than others for a broad range of threshold probabilities in the development cohort (Fig 3C).

## Model validation

### Internal validation
We validated the discrimination of the AS model internally using bootstrapping of 1,000 resamples and the multiple fractional polynomial regression model (Fig S6). The time-dependent AUCs showed results similar to those of the primary analyses.

### External validation
The external validation–independent cohort included 13 patients with a median follow-up time of 22.0 (15.2–32.8) days and a median number of 2.0 (1.0–3.0) measurements; 6 (46.2%) were male (Table S2). The outcome of LD eventually occurred in 7 (53.8%) of these patients, and significant differences in sex were observed. Applying the AS model to the data for 42 serum samples led to the correct assignment 29 (69.0%), 29 (69.0%), and 31 (73.8%) times when the cutoff was defined as 0.3, 0.5, and 0.7, respectively (Fig 4A). The accuracies of the AS model in the external cohort were approximately 70%, similar to the AUCs for the development cohort (Fig 3B).

**Figure 2. Liver dysfunction (LD) associates with a set of dysregulated proteins in the discovery cohort.**
**(A)** Heat map of 20 proteins identified by random forest algorithm in the four grades of LD. **(B)** The AUC assesses the discriminating accuracy of each of the 20 proteins in differentiating LD from COVID-19 patients. **(C)** The association analysis between each of the 20 proteins and LD using the Cox model. **(D)** Correlation analysis between the 20 proteins. LD, liver dysfunction.

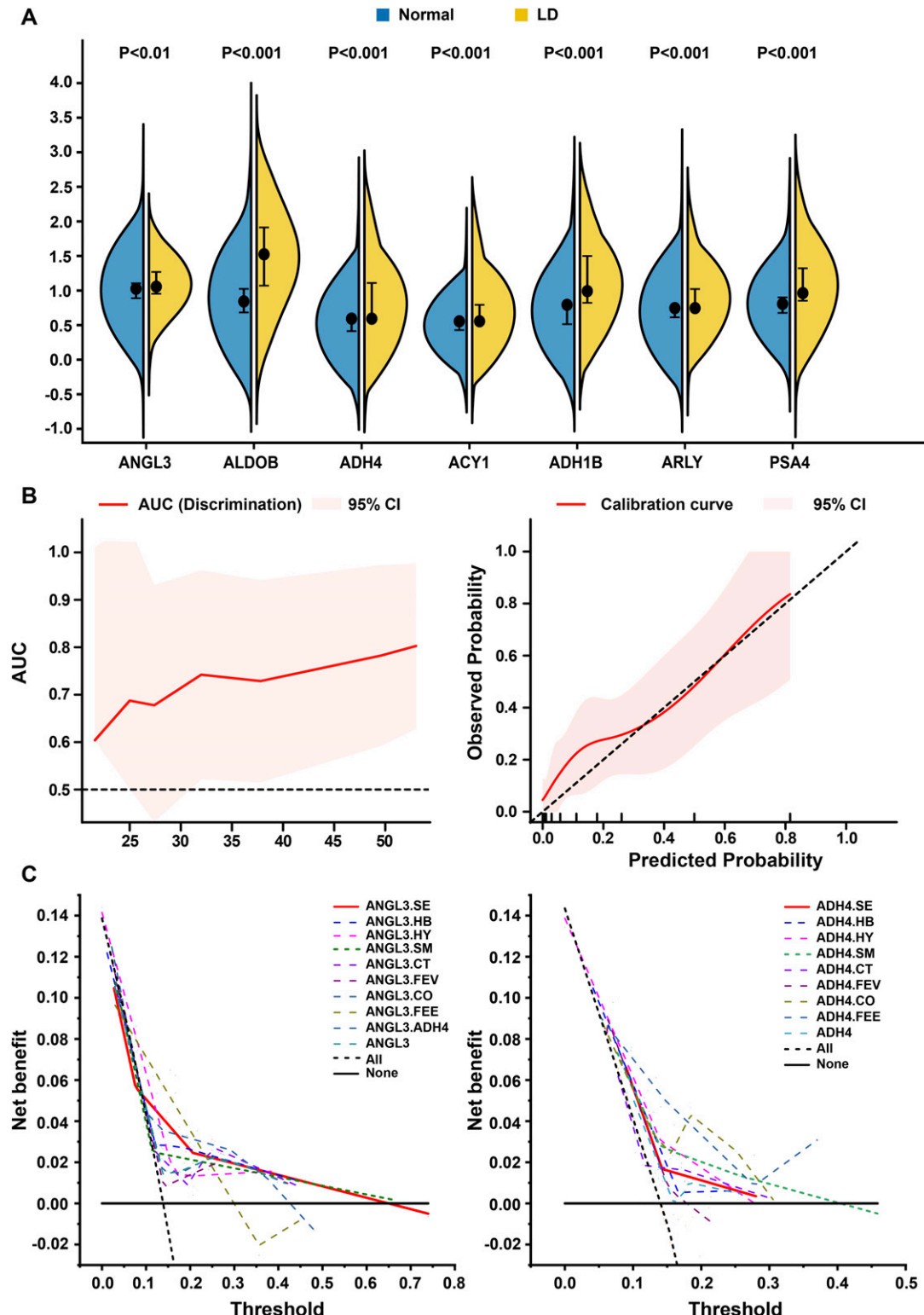

**Figure 3. Differentially expressed proteins between groups in the discovery cohort and the AS model for prediction of COVID-19–related liver dysfunction (LD) in the development cohort.**
**(A)** Violin plot showing the expression values of the seven retained proteins between the normal and LD group. The black dot represents the median value. The error bar represents the interquartile range. **(B)** The discrimination and calibration of the AS model in the development cohort. **(C)** Decision curve analyses for the 18 candidate models in the development cohort. Net benefit is shown with piecewise linear function for each candidate model compared with the treat all and treat none approaches. LD, liver dysfunction; ANGL3.SE, ANGL3.Sex; ANGL3.HB, ANGL3.HBV; ANGL3.HY, ANGL3.Hypertension; ANGL3.SM, ANGL3.Smoking; ANGL3.CT, ANGL3.Chest CT,

### Construction of the nomogram and web-based calculator

Based on the final AS model, a nomogram at 14, 21, 28, 35, and 42 d after hospitalization was constructed using the Cox model (Fig 4B). Fig S7 describes the use of the nomogram to determine the total score and risk of LD in a patient from the development cohort.

A free calculator was established online to enable clinicians to automatically acquire a survival plot and survival probabilities (with 95% CI) of normal liver function in COVID-19 patients by entering values for ANGL3 and sex (https://xixihospital-liufang.shinyapps.io/DynNomapp/) (Fig 4C).

## Discussion

In this study, we developed and validated an AS model and built a free online calculator for the prediction of LD in COVID-19 patients with normal hepatic biochemical parameters on admission. Notably, the AS model only integrates two predictors, namely, a novel biomarker ANGL3 and sex, with maximized generalization and simplicity. The prediction model can be freely accessed at an online web server (https://xixihospital-liufang.shinyapps.io/DynNomapp/).

The predictor ANGL3 obtained among 1,517 proteins is a specific secreted protein originally expressed in the liver, enhancing expression of plasma cholesterol (Gaudet et al, 2017). Interestingly, it has been speculated that cholesterol facilitates SARS-CoV-2 infection by promoting viral spike protein–mediated entry via ACE2 and furin processing (Shoemark et al, 2021). Previous reports have also highlighted the pivotal role of host cholesterol metabolism in the viral life cycle during HCV infection (Foka et al, 2014). Taken together, ANGL3 might indirectly participate in SARS-CoV-2 viral replication through cholesterol metabolism, leading to aggravated LD. The positive association of ANGL3 with LD incidence observed in our study confirms this hypothesis.

The protein ADH4, one of the markers screened by the CoxBoost algorithm, is an enzyme involved in reducing retinaldehyde, which also regulates cholesterol metabolism (Pares et al, 2008). In this study, ADH4-based model calibration did not perform well, potentially because ADH4 is not a liver-specific enzyme and is expressed in many epithelial tissues of the body, such as the blood vessels, skin, cornea, and gastrointestinal mucosa (Pares et al, 2008).

Another predictor of the AS model was sex. Previous studies have reported that males are more prone to LD (Hundt et al, 2020; Wang et al, 2020), consistent with our results (Fig S3), probably resulting from smoking and alcohol use. Moreover, the net benefit comparison between the AS and the ANGL3.SM model showed comparable performance above a threshold of 0.35, suggesting that smoking might be a critical factor in LD.

In the present study, the AS model development followed the principle of at least 10 events per variable (Mallett et al, 2010; Peduzzi et al, 1996) to reduce the risk of overfitting caused by the small sample size of our cohort. With few exceptions, most published COVID-19 prediction models lack comprehensive evaluation of calibration, clinical utility, and external applicability (Wynants et al, 2020). Our study followed the TRIPOD guidelines for model development and evaluated all three aspects mentioned above. We also retained ANGL3 as a continuous variable without arbitrary categorization to avoid loss of information (Gupta et al, 2021). Furthermore, time-dependent discrimination of the AS model provides a dynamic reference for clinicians to predict the appropriate incidence of LD for clinical decision-making.

Our study has several limitations. First, as a single-center and small sample size study, more extensive multicenter clinical research works are required to further validate the potential of the AS model for predicting COVID-19–related LD. Second, although the AS model–based nomogram and online calculator can identify individuals at high risk of LD among COVID-19 patients with normal hepatic biochemical parameters, our study cannot ascertain which treatment strategy might improve the outcomes of these patients. Third, to conclude, the simple AS model presented in this study can predict the incidence of LD earlier than the routine biochemical parameters alanine aminotransferase, aspartate aminotransferase, γ-glutamyl transferase, and total bilirubin. The nomogram and free web-based calculator make the prediction model easily accessible. COVID-19 patients with normal hepatic biochemistry might benefit from early prediction of the probability of subsequent LD to inform clinical decision-making.

## Materials and Methods

### Participants and outcome definitions

The retrospective study cohort was recruited from the Affiliated Hangzhou Xixi Hospital, Zhejiang University School of Medicine (Zhejiang, Southeast China; simplified as Xixi Hospital) from January 2020 to April 2020. Proteomic experiments were conducted among 660 serum samples from 109 COVID-19 patients at the Key Laboratory of Structural Biology of Zhejiang Province, School of Life Sciences, Westlake University (Zhejiang, Southeast China). All serum samples were collected concurrently with clinical blood examination by nurses and subsequently stored at –80°C by two laboratory technicians. Demographic, clinical, and outcome data were collected from the Data Center Laboratory of Xixi Hospital.

Patients with PCR-confirmed positive SARS-CoV-2 RNA (according to the test methods described previously) (Liu et al, 2020) were screened for eligibility. This study is reported following TRIPOD guidance (Collins et al, 2015); it was approved by the Institutional Review Board of Xixi Hospital. Informed consent was waived because of the retrospective nature of this study.

The primary outcome was the subsequent LD after 7 d of admission. Definitions and diagnostic criteria of normal liver function, LD, and its grade (mild, moderate, severe) are summarized in Table S3 (Cai et al, 2020; Liu et al, 2021).

---

ANGL3.FEV, ANGL3.Fever; ANGL3.CO, ANGL3.Cough; ANGL3.FEE, ANGL3.Feebleness; ADH4.SE, ADH4.Sex; ADH4.HB, ADH4.HBV; ADH4.HY, ADH4.Hypertension; ADH4.SM, ADH4.Smoking; ADH4.CT, ADH4.Chest CT; ADH4.FEV, ADH4.Fever; ADH4.CO, ADH4.Cough; ADH4.FEE, ADH4.Feebleness.

**Table 1.  Characteristic of the candidate predictors considered during CoxBoost variable selection among COVID-19 patients with normal hepatic biochemical parameters at baseline.**

| Characteristic | Total | Liver dysfunction | | |
| --- | --- | --- | --- | --- |
| | | No | Yes | P-value |
| No. | 74 | 44 (59.5%) | 30 (40.5%) | |
| Sex | | | | 0.116 |
| Male | 29 (39.2%) | 14 (31.8%) | 15 (50.0%) | |
| Female | 45 (60.8%) | 30 (68.2%) | 15 (50.0%) | |
| Age (years) | 36.0 (26.0–47.8) | 36.5 (25.0–51.0) | 34.5 (29.0–46.8) | 0.864 |
| BMI (kg/m$^2$) | 21.4 (20.3–24.3) | 21.1 (19.7–23.6) | 22.5 (20.7–25.3) | 0.061 |
| Smoking | 6 (8.1%) | 4 (9.1%) | 2 (6.7%) | 0.708 |
| Highest temperature | 37.5 (36.9–38.0) | 37.6 (36.9–38.0) | 37.5 (37.0–38.0) | 0.697 |
| SBP | 126.5 (115.2–135.0) | 121.0 (115.0–134.2) | 130.0 (117.2–135.8) | 0.709 |
| DBP | 76.0 (69.0–85.8) | 76.0 (69.0–87.0) | 76.5 (70.0–84.8) | 0.786 |
| Medication days* | 4.0 (1.0–6.0) | 4.5 (1.0–6.0) | 4.0 (1.0–6.0) | 0.603 |
| Comorbidities | | | | |
| HBV | 2 (2.7%) | 2 (4.5%) | 0 (0.0%) | 0.236 |
| Hypertension | 5 (6.8%) | 5 (11.4%) | 0 (0.0%) | 0.056 |
| CVD | 2 (2.7%) | 2 (4.5%) | 0 (0.0%) | 0.236 |
| Tumor | 1 (1.4%) | 0 (0.0%) | 1 (3.3%) | 0.223 |
| HIV | 0 (0%) | 0 (0%) | 0 (0%) | |
| HCV | 0 (0%) | 0 (0%) | 0 (0%) | |
| COPD | 0 (0%) | 0 (0%) | 0 (0%) | |
| Diabetes | 2 (2.7%) | 2 (4.5%) | 0 (0.0%) | 0.236 |
| Symptoms | | | | |
| Cough | 41 (55.4%) | 23 (52.3%) | 18 (60.0%) | 0.511 |
| Rhinorrhea | 3 (4.1%) | 3 (6.8%) | 0 (0.0%) | 0.144 |
| Fever | 44 (59.5%) | 29 (65.9%) | 15 (50.0%) | 0.171 |
| Diarrhea | 5 (6.8%) | 3 (6.8%) | 2 (6.7%) | 0.98 |
| Rigor | 0 (0%) | 0 (0%) | 0 (0%) | |
| Nausea | 0 (0%) | 0 (0%) | 0 (0%) | |
| Dyspnea | 3 (4.1%) | 2 (4.5%) | 1 (3.3%) | 0.795 |
| Muscular soreness | 3 (4.1%) | 1 (2.3%) | 2 (6.7%) | 0.347 |
| Feebleness | 9 (12.2%) | 3 (6.8%) | 6 (20.0%) | 0.089 |
| Headache | 9 (12.2%) | 6 (13.6%) | 3 (10.0%) | 0.638 |
| Chest congestion | 6 (8.1%) | 5 (11.4%) | 1 (3.3%) | 0.214 |
| Sore throat | 21 (28.4%) | 11 (25.0%) | 10 (33.3%) | 0.435 |
| Sputum | 18 (24.3%) | 11 (25.0%) | 7 (23.3%) | 0.87 |
| Chest CT | | | | 0.236 |
| Single pneumonia | 28 (37.8%) | 18 (40.9%) | 10 (33.3%) | |
| Double pneumonia | 33 (44.6%) | 21 (47.7%) | 12 (40.0%) | |
| Clinical classification | | | | 0.664 |
| Asymptomatic | 1 (1.4%) | 0 (0.0%) | 1 (3.3%) | |
| Mild | 15 (20.3%) | 9 (20.5%) | 6 (20.0%) | |
| Moderate | 56 (75.7%) | 34 (77.3%) | 22 (73.3%) | |

**Table 1. Continued**

| Characteristic | Total | Liver dysfunction | | |
| --- | --- | --- | --- | --- |
| | | **No** | **Yes** | **P-value** |
| Severe | 2 (2.7%) | 1 (2.3%) | 1 (3.3%) | |
| Measurements[#] | | | | |
| Total | 294 | 155 (52.7%) | 139 (47.3%) | |
| Per patient | 3.0 (1.2–4.8) | 2.0 (1.0–4.0) | 3.0 (2.0–5.0) | 0.695 |
| Follow-up time | 18.5 (10.0–32.0) | 16.0 (9.5–31.0) | 22.0 (11.0–32.0) | 0.108 |

Data are n (%) or median (interquartile range) unless otherwise indicated. The asterisk (*) represents the medication days of COVID-19 patients within 7 d of admission. The pound (#) represents the test times of hepatic biochemical parameters in total. BMI, body mass index; COPD, chronic obstructive pulmonary disease; CVD, cardiovascular disease; DBP, diastolic blood pressure; SBP, systolic blood pressure.

## Proteome analysis

### Proteome experiment

Serum samples were inactivated and sterilized at 56°C for 30 min and processed as in the previous study with some modifications (Shen et al, 2020). Briefly, 4 $\mu$l serum from each sample was depleted using High Select Top-14 Abundant Protein Depletion MiniSpin Columns (Thermo Fisher Scientific). Then elutes were denatured, reduced, and alkylated to derive protein lysates. The solutions were diluted with 200 $\mu$l 100 mM triethylammonium bicarbonate followed by a double-step trypsinization (enzyme-to-substrate ratio kept at 1:40 in each step). 10% TFA was added to each sample to quench the enzymatic reaction. Digested peptides were cleaned up with SOLA$\mu$ (Thermo Fisher Scientific) according to the manufacturer's instructions and labeled by TMTpro 16plex reagents (Thermo Fisher Scientific). The reagents were all MS grade. 660 samples in total (633 serum samples for experiment and 27 randomly selected serum samples for quality control) were divided into 44 batches for the labeling experiment, each batch containing 15 samples and one pool. The labeled samples in each batch were combined and fractionated. Each sample was separated into 30 fractions and then consolidated into 10 fractions; dried and re-dissolved fractions with 2% acetonitrile (ACN)/0.1% formic acid of MS grade. The re-dissolved peptides were analyzed with a U3000 HPLC system coupled to an Orbitrap Exploris 480 (Thermo Fisher Scientific) in data-dependent acquisition mode combined to a front-end high-field asymmetric waveform ion mobility spectrometry (FAIMS). Peptides of each fraction were loaded onto a pre-column (3 $\mu$m, 100 Å, 20 mm × 75 mm i.d.) and separated with a 75 min LC gradient at a flow rate of 300 nl/min (analytical column: 1.9 mm, 120 Å, 150 mm × 75 mm i.d.; buffer A: 2% ACN, 98% $H_2O$ containing 0.1% FA; buffer B: 98% ACN, 2% $H_2O$ containing 0.1% FA). FAIMS was operated at two different CVs, –48 and –68 V, respectively. For MS acquisition, the RF level was set at 50% and ion transfer tube temperature at 320°C. The turbo-TMT mode was enabled. The scan range of MS1 was 350–1,800 m/z. Resolutions were set to 60,000 for MS1 and 30,000 for MS2. Normalized AGC target was set at 300% for MS1 and 200% for MS2. The maximum injection time was set as 50 ms for MS1 and 86 ms for MS2. Dynamic exclusion was on, and mass tolerance was set to ±10 ppm. Intensity threshold was set at 20,000 for MS1, and HCD collision energy was fixed to 38%. MS/MS data were recorded in centroid mode. Isolation window was set to 0.7 m/z.

### Proteomic data analysis

MS data for proteomics were processed with Proteome Discoverer (version 2.4.0.305; Thermo Fisher Scientific) against a manually annotated and reviewed *Homo sapiens* protein FASTA database (Swiss-Prot, 27 April 2020). For parameter settings, enzyme digestion was set to full-specific trypsin with two missed cleavages. Static modifications were TMTpro of lysine residues and N-terminus peptides and carbamidomethylation of cysteine. Dynamic modifications were set to oxidation of methionine and acetylation of N-terminus peptides. Mass tolerance for precursors and product ions was fixed at 10 ppm and 0.02 Da, respectively. The peptide–spectrum match allowed 1% target false discovery rate (strict) and 5% target false discovery rate (relaxed). 1,517 proteins in total were quantified. The protein abundance ratio of samples to the pooled sample within each batch was considered as the relative protein abundance ratio for the subsequent analysis.

### Quality control of proteome data

Multiple quality control of proteome data was performed as previously stated (Shen et al, 2020). Briefly, mouse liver digest was used as a standard sample for instrument performance evaluation. To avoid carry-over, a blank sample (buffer A) was run between every four sample injections. A pooled peptide sample labeled by TMT pro-126 was contained in each batch for aligning data from different batches and evaluation of quantitative accuracy. We first evaluated the quantitative ratio distribution of 633 samples. The outliers (above upper quartile) were imputed as two times the interquartile. Then we checked the correlation of 27 technical replicates by Pearson correlation coefficient (Fig S8A). R package limma was used to remove batch effects from experiments and instruments. There were no significant differences between 44 batches using principal component analysis (Fig S8B).

## Candidate protein selection

Ninety-eight of the 109 patients with LD outcomes were enrolled as the discovery cohort for identifying biomarkers. We estimated the MDA for each of the 1,517 quantified proteins by the RF algorithm in the discovery cohort and prioritized 20 proteins with an MDA larger than 3 for subsequent analysis (Fig S1). In RF analysis, 500 trees were built with 10-fold cross-validation, and this was repeated 100 times. We performed univariable and multivariate Cox models, including age, sex, and BMI, and AUCs (equal to the concordance

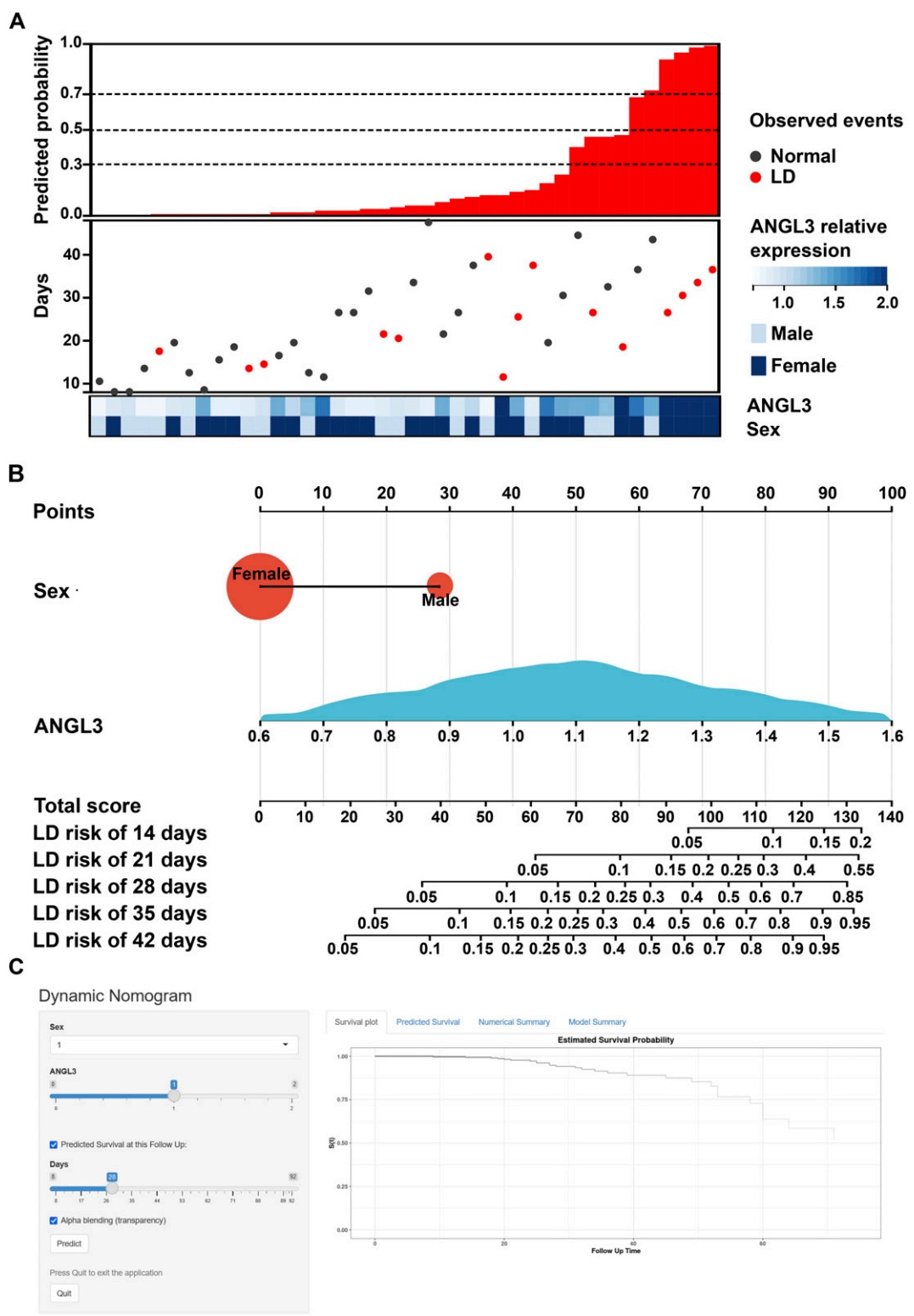

**Figure 4. External validation, nomogram, and online web-based calculator of AS model.**
**(A)** The external discrimination validation of the AS model using another dataset from Taizhou hospital. Predicted probability comes from the AS model. Observed events represent the actual occurrences of LD in the external cohort. **(B)** Characteristics in the nomogram to predict the risk of liver dysfunction (LD) in COVID-19 patients with normal liver biochemical parameters on admission. Patient predictive values are located on the axis of each variable; a line is then drawn upwards at a 90° angle to determine the number of points for that particular variable. The sum of these numbers is located on the total score axis, and a line is drawn at a 90° angle downward to

statistics) to estimate the ability of each protein to discriminate LD versus normal (Fig 2B and C). A protein–protein correlation plot was constructed for the 20 proteins (Fig 2D). Some strong-correlation proteins were eliminated from candidate variates to minimize potential collinearity and overfitting bias. ANGL3, ALDOB, ADH4, ACY1, ADH1B, ARLY, and PSA4 were eventually retained (Fig 3A).

### Selection of potential clinical predictors and model development

We included multiple clinical characteristics in the model development, including demographic variables, medical history, symptoms, and chest CT results on hospital admission. The demographic variables examined included age, sex, BMI, smoking status, highest temperature, systolic blood pressure, and diastolic blood pressure. Medical history included medications, HBV, hypertension, cardiovascular disease, tumor, HIV, HCV, chronic obstructive pulmonary disease, and diabetes. We also considered symptoms, such as cough, rhinorrhea, fever, pidiarrhea, rigor, nausea, dyspnea, muscular soreness, feebleness, headache, chest congestion, sore throat, and sputum. Chest CT results included normal, single pneumonia, and double pneumonia.

Of 98 individuals in the discovery cohort, 74 patients hospitalized with normal liver function for 7 d were selected as the development cohort. Peak values of ANGL3, ALDOB, ADH4, ACY1, ADH1B, ARLY, and PSA4 within 7 d of admission to the hospital were subject to next-stage selection together with the other 31 clinical variables, as described above.

Given that time may affect the discrimination of the prediction model, the CoxBoost algorithm (an XGBoost-based survival estimator model) (Unterhuber et al, 2021) was applied to select the most potent predictors that could be retained in the final model. To maximize the generalizability and simplicity of the prediction model, we restricted the number of variables to 2, and the combination was set to contain at least one protein marker. Each combination was introduced into a Cox regression model in view of the long-term horizon for predictions (during inpatient and non-inpatient periods) to quantify AUC values for given times.

### Model selection

During selection, the discrimination (ability to distinguish individuals who will develop LD from those who will not, as quantified as the AUC), calibration (consistency between the predicted and observed probability, as evaluated by a calibration plot), and clinical utility (as quantified by the net benefit of a decision curve) were assessed among the above-combined prediction models (Steyerberg & Vergouwe 2014). An ideal calibration slope is 1, which suggests that the risk of observed outcomes matches the risk predicted. Decision curve analysis permits evaluation of clinical utility by quantifying the trade-off between correct discrimination of true positives and incorrect discrimination of false positives, as weighted based on the threshold probability (Vickers et al, 2019). The threshold probability represents the benefit ratio for the intervention. In this article, decision curve analysis was conducted at 28 d after hospital admission to quantify the net benefit of implementing the model in clinical practice. For all of the decision curves, data were fitted with a piecewise linear function among all replicates with the following algorithm: double $yi_1$, $yi_2$;

$$yi_1 = a_1 + k_1 xi_1 ;$$

$$yi_2 = yi_1 + k_2 \ (xi_2 - xi_1);$$

$$if \ (x < xi_1)$$

$$y = a_1 + k_1 x$$

$$else \ if \ (x < xi_2)$$

$$y = yi_1 + k_2 \ (x - xi_1);$$

$$else$$

$$y = yi_2 + k_3 \ (x - xi_2);$$

Finally, the model that performed comprehensively best across discrimination, calibration, and clinical utility was used to construct a dynamic nomogram plot, which was then used to build a free online calculator.

### Model validation

We utilized bootstrapping of 1,000 resamples (with replacement) and multiple fractional polynomial regression modeling to internally validate the stability of the final model using complete development case data.

To further validate the generalizability of the AS model, we collected 42 serum samples from 13 COVID-19 patients with normal hepatic biochemical parameters at Taizhou Hospital of Zhejiang province, affiliated to Wenzhou Medical University between January 2020 and February 2020 at the time of hospitalization for the external validation cohort. The sex information and the maximum values of ANGL3 within 7 d of hospital admission were selected for validation.

A result was considered statistically significant when the two-tailed $P$-value was below 0.05. R software versions 3.6.3 and 4.0.5 (www.r-project.org) were used for the statistical analyses.

# Data Availability Statement

The original contributions presented in the study are included in the article/Supplementary Material. Further inquiries can be directed to the corresponding authors.

# Supplementary Information

---

the LD risk axes. **(C)** The free online tool for identifying the patients with low risk of liver dysfunction (LD) in COVID-19 patients. Male was represented at an arbitrary value of 1 (female = 2). ANGL3 represents the relative expression during proteomics analysis. LD, liver dysfunction.

# Acknowledgements

We thank Westlake University Supercomputer Center for assistance in data generation and storage and the Mass Spectrometry & Metabolomics Core Facility at the Center for Biomedical Research Core Facilities of Westlake University for sample analysis. The authors disclosed receipt of the following financial support for the research, authorship, and publication of this article: Natural Science Foundation of Zhejiang Province (LQ22H100001), Medical and Health Research Project of Zhejiang Province (2022508831), the Hepatology (Traditional Chinese and Western Medicine) of Hangzhou Medical Peak Subject, Science and Technology Development Program of Hangzhou (202004A20), Hangzhou Pharmaceutical Health and Technology Project (Z20220098), the National Key R&D Program of China (2020YFE0202200), the National Natural Science Foundation of China (81972492, 21904107), Zhejiang Provincial Natural Science Foundation for Distinguished Young Scholars (LR19C050001), and Westlake Education Foundation, Tencent Foundation (2020).

## Author Contributions

J Bao: conceptualization, funding acquisition, project administration, and writing—original draft, review, and editing.
S Liu: supervision.
X Liang: methodology and writing—review and editing.
C Wang: data curation.
L Cao: data curation.
Z Li: data curation.
F Wei: data curation.
A Fu: data curation.
Y Shi: data curation.
B Shen: data curation.
X Zhu: data curation.
Y Zhao: data curation.
H Liu: data curation.
L Miao: data curation.
Y Wang: data curation.
S Liang: data curation.
L Wu: data curation.
J Huang: supervision and funding acquisition.
T Guo: supervision, funding acquisition, methodology, and writing—review and editing.
F Liu: conceptualization, resources, data curation, formal analysis, funding acquisition, validation, visualization, methodology, project administration, and writing—original draft, review, and editing.

## Conflict of Interest Statement

The authors declare that they have no conflicts of interest.

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
