## [Reviewer comments · Life Science Alliance]

Life Science Alliance

A simple prediction model for COVID-19 liver dysfunction in patients with normal hepatic biochemical

Jianfeng Bao, Shourong Liu, Xiao Liang, Congcong Wang, Lili Cao, Zhaoyi Li, Furong Wei, Ai Fu, Yingqiu Shi, Bo Shen, Xiaoli Zhu, Yuge Zhao, Hong Liu, Liangbin Miao, Yi Wang, Shuang Liang, Linyan Wu, Jinsong Huang, Tiannan Guo, and Fang Liu
DOI: <https://doi.org/10.26508/lsa.202201576>

Corresponding author(s): Fang Liu, Insitute of Hepatology and Epidemiology, Affiliated Hangzhou Xixi Hospital, Zhejiang University School of Medicine; Tiannan Guo, Westlake University; and Jinsong Huang, Department of Hepatology, Affiliated Hangzhou Xixi Hospital, Zhejiang University School of Medicine

Review Timeline:

Submission Date:	2022-06-23
Editorial Decision:	2022-08-02
Revision Received:	2022-09-08
Editorial Decision:	2022-09-23
Revision Received:	2022-09-30
Accepted:	2022-09-30

Transaction Report:

August 2, 2022

Re: Life Science Alliance manuscript #LSA-2022-01576-T

Fang Liu
Institute of Hepatology and Epidemiology, Affiliated Hangzhou Xixi Hospital, Zhejiang University School of Medicine
Xihu District, 2 Hengbu Road
Hangzhou, Zhejiang 310023
China

Dear Dr. Liu,

Thank you for submitting your manuscript entitled "A simple prediction model for COVID-19 liver dysfunction in patients with normal hepatic biochemical" to Life Science Alliance. The manuscript was assessed by an expert reviewer, whose comments are appended to this letter. We invite you to submit a revised manuscript addressing the Reviewer comments.

When submitting the revision, please include a letter addressing the reviewer's comments point by point.

Thank you for this interesting contribution to Life Science Alliance. We are looking forward to receiving your revised manuscript.

Sincerely,

B. MANUSCRIPT ORGANIZATION AND FORMATTING:

Reviewer #1 (Comments to the Authors (Required)):

Comments to the MS: A prediction model for COVID-19 liver dysfunction in patients with normal 2 hepatic biochemical (MS ID: LSA-2022-01576-T)

The pandemic of COVID-19 in the past 3 years have posed multiple challenges in medical practice, and may previously unidentified disorders have emerged. In the field of hepatology, COVID-19 virus has been shown to cause liver injury leading to severe clinical consequences. Indeed, various level of liver damage has been reported in ~50% of COVID patients. In addition, some hepatotoxic-antiviral drugs that were used to treat COVID may also cause liver injury leading to liver failure. Commonly used indexes for liver functions may not be sensitive enough to predict or diagnose liver injury. Thus, in these perspectives, it is highly desirable to develop algorithms that can be used to predict liver injury in COVID-19 patients.

In this study, the authors conducted proteomic analysis on the serum samples of 397 cases of COVID-19 patients. They used 74 COVID-19 patients with normal hepatic biochemicals and established 19 simple machine learning models. They validated these models in another independent cohort and found that the predicting accuracy of liver damage could reach 69-74%, with ANGL3 and sex (AS) model being the most reliable and sensitive in predicting the early occurrence of liver injury in COVID infection than the usual biochemicals of liver function tests such as ALT, AST, GGT, and TBIL. The authors have already applied this model in the management of COVID-19 patients. It is anticipated that this predicting model may facilitate the early identification of the patients with potential liver injury and decrease the incidence of non-SARS-CoV-2-related complications.

Major:

1. "Proteomics is a powerful tool to profile the molecular modulation in COVID-19 patients". This statement narrows the application of proteomics analysis. In fact, proteomics can be used to identify the molecular mechanisms of any diseases at the protein level, not just limited to COVID-19.
2. One of the limitations of this study is that the developed model was only validated in 13 patients, and the median follow-up time was only 22 days. The reviewer understands the difficulty in recruiting prospective COVID-19 patients (thus, not required in this ms). However, a validation of this predicting model in a large cohort of prospective COVID-19 patients with much longer follow up would provide more robust evidence of the clinical applicability of this model in clinical practice.
3. In Tables 1, S2 and S3, only male patients were analyzed (29/74 in Table 1, 47/98 in Table S2; 6/13 in Table S3). Why female patients were not included?
4. In the "Details for proteomics analysis", the detailed descriptions for "Proteome experiment analysis" could be put in "supplementary data" if the journal allows, otherwise, could be much more condensed. Some parts, e.g., "....660 samples in total (633 serum samples and 27 technical replicates)" need more clarification: what did the authors mean by 27 technical replicates?
5. In Table S1, some descriptions can be confusing. E.g., the descriptions for "Moderate liver dysfunction" could be improved to AST {greater than or equal to}35 U/L, ALT 41-120 U/L, GGT 46-90 U/L, TBIL 21-41 µmol/L.
6. In Tables 1, S2 and S3, what did the author mean by "Measurements"? needs clarification.
- 7.

Minor:

1. There are considerable number of language errors/inaccuracies in the manuscript. E.g., ".....However, hepatic abnormalities possess many manifestations, including liver congestion, inflammatory response, drug-induced liver damage, and hepatocytes infection (Marjot et al., 2021). This brings to the difficulties of identifying susceptible patients at an early stage". Although the reviewer could manage to understand what the authors were trying to express, the statement would be much clearer with a solid language polish.

2. The authors are advised to undertake a rigorous English editing (e.g., by a professional language service or a native English speaker).

Overall recommendation

Although the study has limitations, it is nevertheless a well-conducted timely study aiming to developing algorithms for identifying COVID-19 patients with potential liver injury, and hence carries with it some important insights to readers. If in the future these models especially the AS model could be validated in the real-world studies, this model would aid the clinical management of

patients with COVID-19 infections. Perhaps the insights within this study may be extrapolated to the patients with other viral infections.

I recommend this manuscript be accepted with some minor revisions.

Editors

1. A letter addressing the reviewers' comments point by point.

Thanks for your reminder. We have revised the manuscript and answered the reviews' comments point by point.

2. An editable version of the final text (.DOC or .DOCX) is needed for copyediting (no PDFs).

Thanks for your reminder. We uploaded the final manuscript in the needed format (.docx).

3. High-resolution figures, supplementary figures, and video files uploaded as individual files: See our detailed guidelines for preparing your production-ready images, <https://www.life-science-alliance.org/authors>

Thanks for your reminder. We have revised all Figures according to the guidelines online.

4. Summary blurb (enter in submission system): A short text summarizing in a single sentence the study (max. 200 characters including spaces). This text is used in conjunction with the titles of papers, hence should be informative and complementary to the title and running title. It should describe the context and significance of the findings for a general readership; it should be written in the present tense and refer to the work in the third person. Author names should not be mentioned.

Thanks for your reminder. The summary blurb "The composite model of sex and the novel protein marker ANGL3 exhibits great potential in predicting the risk of developing LD in COVID-19 patients with normal hepatic biochemicals on admission." has been submitted online.

Reviewers**Major comments:**

The pandemic of COVID-19 in the past 3 years has posed multiple challenges in medical practice, and many previously unidentified disorders have emerged. In the field of hepatology, COVID-19 virus has been shown to cause liver injury leading to severe clinical consequences. Indeed, various level of liver damage has been reported in ~50% of COVID patients. In addition, some hepatotoxic-antiviral drugs that were used to treat COVID may also cause liver injury

leading to liver failure. Commonly used indexes for liver functions may not be sensitive enough to predict or diagnose liver injury. Thus, in these perspectives, it is highly desirable to develop algorithms that can be used to predict liver injury in COVID-19 patients.

In this study, the authors conducted proteomic analysis on the serum samples of 397 cases of COVID-19 patients. They used 74 COVID-19 patients with normal hepatic biochemicals and established 19 simple machine learning models. They validated these models in another independent cohort and found that the predicting accuracy of liver damage could reach 69-74%, with ANGL3 and sex (AS) model being the most reliable and sensitive in predicting the early occurrence of liver injury in COVID infection than the usual biochemicals of liver function tests such as ALT, AST, GGT, and TBIL. The authors have already applied this model in the management of COVID-19 patients. It is anticipated that this predicting model may facilitate the early identification of the patients with potential liver injury and decrease the incidence of non-SARS-CoV-2-related complications.

1. "Proteomics is a powerful tool to profile the molecular modulation in COVID-19 patients". This statement narrows the application of proteomics analysis. In fact, proteomics can be used in identify the molecular mechanisms of any diseases at the protein level, not just limited to COVID-19.

Thanks for your comments. According to the reminder, we have revised the sentence "Proteomics is a powerful tool to profile the molecular modulation in COVID-19 patients" as "Proteomics profiling has the ability to shed light on molecular changes reflected in sera from COVID-19 patients" (Introduction section, line 96-97 of revised manuscript).

2. One of the limitations of this study is that the developed model was only validated in 13 patients, and the median follow-up time was only 22 days. The reviewer understands the difficulty in recruiting prospective COVID-19 patients (thus, not required in this ms). However, a validation of this predicting model in a large cohort of prospective COVID-19 patients with much longer follow up would provide more robust evidence of the clinical applicability of this model in clinical practice.

Thank you very much for your kind understanding. According to your suggestion, the sentence of "First, the procurement of discovery cohort from a single center may limit its generality in other populations, and thus further validations in independent cohorts are warranted." was revised as "First, as a single center and small sample size study, more extensive multicenter clinical researches are required to further validate the potential of the AS model for predicting COVID-19-related LD." (Discussion section, lines 257-259).

3. In Tables 1, S2 and S3, only male patients were analyzed (29/74 in Table 1, 47/98 in Table S2; 6/13 in Table S3). Why female patients were not included?

Thanks for your question. We added the information on females to the abovementioned tables (line 506 of the revised manuscript, lines 96 and 111 of modified supplementary material).

4. In the "Details for proteomics analysis", the detailed descriptions for "Proteome experiment analysis" could be put in "supplementary data" if the journal allows, otherwise, could be much more condensed. Some parts, e.g., "...660 samples in total (633 serum samples and 27 technical

replicates)" need more clarification: what did the authors mean by 27 technical replicates?

Thanks for your question. The technical replicates here refer to the repeated mass spectrometric analysis of 27 randomly selected serum samples, which were used for quality control. We have accordingly put the detailed descriptions for "Proteome Experiment Analysis" in "Supplementary material" and revised the related text, as copied below: 660 samples in total (633 serum samples for the experiment and 27 randomly selected serum samples for quality control) were divided into 44 batches for the labeling experiment, each batch containing fifteen samples and one pool (Supplementary material, lines 42-45).

5. In Table S1, some descriptions can be confusing. E.g., the descriptions for "Moderate liver dysfunction" could be improved to AST {greater than or equal to}35 U/L, ALT 41-120 U/L, GGT 46-90 U/L, TBIL 21-41 $\mu\text{mol/L}$.

Thanks for your suggestion. We modified the description for "Moderate liver dysfunction" as AST ≥ 35 U/L combined with ALT: 40-120 U/L, GGT: 45-90 U/L, or TBIL: 20.52-41.04 $\mu\text{mol/L}$ (Supplementary material, line 125).

6. In Tables 1, S2 and S3, what did the author mean by "Measurements"? needs clarification.

Thanks for your kind reminder. The word "Measurements" here represents the test times of hepatic biochemical parameters in total. We have added the clarification of "Measurements" under Tables 1, S1, and S2 (line 512 of the revised manuscript, lines 99 and 113 of modified supplementary material).

Minor comments:

1. There are considerable number of language errors/inaccuracies in the manuscript. E.g., ".....However, hepatic abnormalities possess many manifestations, including liver congestion, inflammatory response, drug-induced liver damage, and hepatocytes infection (Marjot et al., 2021). This brings to the difficulties of identifying susceptible patients at an early stage". Although the reviewer could manage to understand what the authors were trying to express, the statement would be much clearer with a solid language polish.

2. The authors are advised to undertake a rigorous English editing (e.g., by a professional language service or a native English speaker).

Although the study has limitations, it is nevertheless a well-conducted timely study aiming to developing algorithms for identifying COVID-19 patients with potential liver injury, and hence carries with it some important insights to readers. If in the future these models especially the AS model could be validated in the real-world studies, this model would aid the clinical management of patients with COVID-19 infections. Perhaps the insights within this study may be extrapolated to the patients with other viral infections.

Special thanks to you for your comments about our research. We undertook rigorous English editing by a professional language service, the modified parts have been marked in red in the revised manuscript, and the editing certificate has been uploaded online.

Other changes :

The original Fig S6 and Fig S8 were added majuscule “A” and “B” for discerning.

We tried our best to improve the manuscript and made some changes in the manuscript. These changes will not influence the content and framework of the paper. And here, we listed the changes above and marked them in red in the revised paper. We appreciate for Editors/Reviewers' warm work earnestly and hope that the correction will meet with approval. Once again, thank you very much for your comments and suggestions.

We look forward to hearing from you at your earliest convenience.

Yours sincerely,

Fang Liu

September 23, 2022

RE: Life Science Alliance Manuscript #LSA-2022-01576-TR

Ms. Fang Liu

Institute of Hepatology and Epidemiology, Affiliated Hangzhou Xixi Hospital, Zhejiang University School of Medicine
Xihu District, 2 Hengbu Road
Hangzhou, Zhejiang 310023
China

Dear Dr. Liu,

Thank you for submitting your revised manuscript entitled "A simple prediction model for COVID-19 liver dysfunction in patients with normal hepatic biochemical". We would be happy to publish your paper in Life Science Alliance pending final revisions necessary to meet our formatting guidelines.

- please add ORCID ID for both corresponding authors and note that LSA allows a maximum of two corresponding authors
- please add the Twitter handle of your host institute/organization as well as your own or/and one of the authors in our system
- please add your supplementary figure legends to the main manuscript text
- please add a callout for Figure S8 to your main manuscript text
- The Methods and Reference in the Supplemental Material file should be incorporated into the Materials and Methods and Reference sections of the main text. The legends for the Supplemental Figures should be added at the end of the Figure Legend section of the main file. This supplemental file is then unnecessary.

A. FINAL FILES:

B. MANUSCRIPT ORGANIZATION AND FORMATTING:

Sincerely,

Reviewer #1 (Comments to the Authors (Required)):

The authors have satisfactorily addressed all comments raised by the reviewers.
I have no further comments to make for this ms.
I am happy for the ms be accepted for publication in LSA.

Dear Professor Eric Sawey:

Thanks for your suggestions concerning our manuscript entitled "**A simple prediction model for COVID-19 liver dysfunction in patients with normal hepatic biochemical**" (ID: LSA-2022-01576-T). We have studied the suggestions carefully and made corrections. The revised portion is marked in red on the paper. The main corrections in the paper are as follows:

-please add ORCID ID for both corresponding authors and note that LSA allows a maximum of two corresponding authors

-please add the Twitter handle of your host institute/organization as well as your own or/and one of the authors in our system

-please add your supplementary figure legends to the main manuscript text

-please add a callout for Figure S8 to your main manuscript text

-The Methods and Reference in the Supplemental Material file should be incorporated into the Materials and Methods and Reference sections of the main text. The legends for the Supplemental Figures should be added at the end of the Figure Legend section of the main file. This supplemental file is then unnecessary.

Thank you very much for your permit regarding our requirement to list three corresponding authors for this present collaborative study, and we have revised the manuscript according to the other suggestions point by point.

We tried to add ORCID for both corresponding authors, but there is no operational button of online system for us to do this, such as the following image:

Author #	Corr Author	Name	Email	Organization
19	[ ]	Tiannan Guo	g*****@westlake.edu.cn	Westlake University

ORCID  : N/A

* Person Title:

* Name:

First Middle Last

* Email:

* Institution:

The ORCID IDs of Jinsong Huang and Tiannan Guo were 0000-0003-1254-1181 and 0000-0003-3869-7651, respectively. Can we respectfully request you to help us for this issue?

2. An editable version of the final text (.DOC or .DOCX) is needed for copyediting (no PDFs).

Thanks for your reminder. We uploaded the final manuscript in the needed format (.docx).

3. High-resolution figures, supplementary figures, and video files uploaded as individual files: See our detailed guidelines for preparing your production-ready images, <https://www.life-science-alliance.org/authors>

Thanks for your reminder. We have revised all Figures according to the guidelines online.

4. Summary blurb (enter in submission system): A short text summarizing in a single sentence the study (max. 200 characters including spaces). This text is used in conjunction with the titles of papers, hence should be informative and complementary to the title and running title. It should describe the context and significance of the findings for a general readership; it should be written in the present tense and refer to the work in the third person. Author names should not be mentioned.

Thanks for your reminder. The summary blurb "The composite model of sex and the novel protein marker ANGL3 exhibits great potential in predicting the risk of developing LD in COVID-19 patients with normal hepatic biochemicals on admission." has been submitted online.

We tried our best to improve the manuscript and made some changes in the manuscript. These changes will not influence the content and framework of the paper. And here, we listed the changes above and marked them in red in the revised paper. We appreciate for Editors/Reviewers' warm work earnestly and hope that the correction will meet with approval. Once again, thank you very much for your comments and suggestions.

We look forward to hearing from you at your earliest convenience.

Yours sincerely,

Fang Liu

September 30, 2022

RE: Life Science Alliance Manuscript #LSA-2022-01576-TRR

Ms. Fang Liu
Institute of Hepatology and Epidemiology, Affiliated Hangzhou Xixi Hospital, Zhejiang University School of Medicine
Xihu District, 2 Hengbu Road
Hangzhou, Zhejiang 310023
China

Dear Dr. Liu,

Thank you for submitting your Research Article entitled "A simple prediction model for COVID-19 liver dysfunction in patients with normal hepatic biochemical". It is a pleasure to let you know that your manuscript is now accepted for publication in Life Science Alliance. Congratulations on this interesting work.

DISTRIBUTION OF MATERIALS:

Again, congratulations on a very nice paper. I hope you found the review process to be constructive and are pleased with how the manuscript was handled editorially. We look forward to future exciting submissions from your lab.

Sincerely,
